# Applications of Ketogenic Diets in Patients with Headache: Clinical Recommendations

**DOI:** 10.3390/nu13072307

**Published:** 2021-07-05

**Authors:** Cherubino Di Lorenzo, Giovanna Ballerini, Piero Barbanti, Andrea Bernardini, Giacomo D’Arrigo, Gabriella Egeo, Fabio Frediani, Riccardo Garbo, Giulia Pierangeli, Maria Pia Prudenzano, Nicoletta Rebaudengo, Grazia Semeraro, Giulio Sirianni, Mariarosaria Valente, Gianluca Coppola, Mackenzie C. Cervenka, Giovanni Spera

**Affiliations:** 1Department of Medico-Surgical Sciences and Biotechnologies, Sapienza University of Rome Polo Pontino, 04100 Latina, Italy; gianluca.coppola@uniroma1.it; 2Multidisciplinary Center for Pain Therapy, Piero Palagi Hospital, USL Toscana Centro, 50122 Florence, Italy; giovannaballerini@icloud.com; 3Headache and Pain Unit, IRCCS San Raffaele Pisana, 00163 Rome, Italy; piero.barbanti@sanraffaele.it (P.B.); gabriella.egeo@sanraffaele.it (G.E.); 4Department of Neuroscience and Rehabilitation, San Raffaele University, 00163 Rome, Italy; 5Clinical Neurology Unit, Misericordia University Hospital, Santa Maria Della Misericordia University Hospital, 33100 Udine, Italy; bernardini.an@gmail.com (A.B.); riccardo.garbo@outlook.it (R.G.); mariarosaria.valente@uniud.it (M.V.); 6Headache Center, Neurology & Stroke Unit, San Carlo Borromeo Hospital, ASST Santi Paolo e Carlo, 20142 Milan, Italy; giacomo.darrigo@gmail.com (G.D.); fabio.frediani@asst-santipaolocarlo.it (F.F.); 7IRCCS Istituto Delle Scienze Neurologiche di Bologna, 40139 Bologna, Italy; giulia.pierangeli@unibo.it; 8Department of Biomedical and NeuroMotor Sciences, University of Bologna, 40127 Bologna, Italy; 9Headache Center, Department of Basic Medical Sciences, Neurosciences and Sense Organs, University of Bari, 70124 Bari, Italy; mariapia.prudenzano@virgilio.it; 10Neurology Service, Humanitas Gradenigo, 10153 Turin, Italy; nicoletta.rebaudengo@gradenigo.it; 11Associazione Eupraxia, Dietary Section, 00171 Rome, Italy; grazia_smrr@libero.it (G.S.); giulio.sirianni@gmail.com (G.S.); 12Neurology Unit, Department of Medicine (DAME), University of Udine, Piazzale Santa Maria Della Misericordia 15, 33100 Udine, Italy; 13Department of Neurology, Johns Hopkins University School of Medicine, Baltimore, MD 21287, USA; mcerven1@jhmi.edu; 14Department of Experimental Medicine, Section of Medical Pathophysiology, Food Science and Endocrinology, Sapienza University of Rome, 00161 Rome, Italy; giannispera@yahoo.com

**Keywords:** ketogenic diets, ketosis, ketones, clinical recommendations, headache, migraine, cluster headache

## Abstract

Headaches are among the most prevalent and disabling neurologic disorders and there are several unmet needs as current pharmacological options are inadequate in treating patients with chronic headache, and a growing interest focuses on nutritional approaches as non-pharmacological treatments. Among these, the largest body of evidence supports the use of the ketogenic diet (KD). Exactly 100 years ago, KD was first used to treat drug-resistant epilepsy, but subsequent applications of this diet also involved other neurological disorders. Evidence of KD effectiveness in migraine emerged in 1928, but in the last several year’s different groups of researchers and clinicians began utilizing this therapeutic option to treat patients with drug-resistant migraine, cluster headache, and/or headache comorbid with metabolic syndrome. Here we describe the existing evidence supporting the potential benefits of KDs in the management of headaches, explore the potential mechanisms of action involved in the efficacy in-depth, and synthesize results of working meetings of an Italian panel of experts on this topic. The aim of the working group was to create a clinical recommendation on indications and optimal clinical practice to treat patients with headaches using KDs. The results we present here are designed to advance the knowledge and application of KDs in the treatment of headaches.

## 1. Introduction

Headaches are one of the most common symptoms reported by people during their lifespan, with an estimated prevalence of 47% [1]. The international classification of headache disorders, 3rd edition (ICHD-3) [2], classifies different primary and secondary forms of headache. The most frequent forms are tension-type headache (22%) and migraine (15%), respectively the second and third most prevalent disorders worldwide, after dental caries [3]. Additionally, in terms of disability, the headache is the third most disabling condition, as measured in years of life lived with disability [3]. Despite the global burden of disease, according to the World Health Organization, headache disorders remain underestimated and undertreated worldwide [1]. This undertreatment of headache is most significant for patients with migraines who require preventive therapy to reduce the number of monthly attacks, and was recently confirmed in large epidemiological and regional studies [4,5]. Preventive treatment is recommended for patients with at least 4 days of migraine headache per month, or who find symptomatic treatments ineffective or not well tolerated [6]. It was estimated that 38.8% of subjects with migraines are eligible for preventive treatment, but just 12.4% use them [7]. The goal of pharmacological prophylaxis for migraine is to decrease the frequency, severity, and duration of each attack, increase responsiveness to symptomatic treatments and improve the patient’s quality of life [6]. Unfortunately, all the drugs commonly used to prevent migraines were developed originally for other purposes. Hence, most of the preventive treatments are characterized by a relatively low responder rate (so, patients will require more preventive medications concomitantly), potentially dangerous drug–drug interactions, and several side effects that result in high rates of switching or discontinuing treatment [8]. The most recent biological treatments (OnabotulinumtoxinA and monoclonal antibodies against calcitonin-gene-related peptide (CGRP)) showed a better profile of safety, but they are not indicated for all patients and concerns have been raised about cost/effectiveness [9]. Moreover, these innovative treatments are not effective in all patients that try them. In summary, the outcome of preventive pharmacological treatments, when started, is unsatisfactory for many patients with migraine, leading to the possible worsening of headache by the development of complications, such as acute medication overuse and chronic daily headache (CDH).

There is a growing interest in non-pharmacological approaches (NPAs) to headache management. American Headache Association guidelines [10] recommend behavioral therapy, relaxation therapy [11], and other lifestyle changes [12,13] as the cornerstones of headache prophylaxis and recommend discussing these among the treatment options with all patients [10]. Among the NPAs, other than relaxation and lifestyle changes, the greatest interest is in nutraceuticals [14], and nutritional interventions [15], including food selection (in terms of potential food trigger avoidance) [16,17], and potential dietary treatment [18,19]. Here, we explore the ketogenic diet (KD) as a promising dietary treatment approach [20,21,22], used for one century is used to treat epilepsy and other neurological conditions.

This paper summarizes the existing evidence on the potential benefits of KD on headaches and synthesized results of a working group composed of Italian experts in the treatment of patients with headaches using KDs. The panel defined common indications and practices to address the use of KDs for headaches, in an effort to standardize the treatment and research protocols by creating a clinical recommendation useful for other neurologists, nutrition specialists, clinicians of other specialties, general practitioners, and researchers.

The summary of our study group recommendations represents expert opinions in the field of headache management, based on members’ personal clinical experiences, in instances where lacking rigorous scientific studies to create evidence-based guidelines.

## 2. Ketogenic Diet: Overview

KD, also referred to as “simulated fasting”, was developed around 1920 as an attempt to prolong the benefits of fasting to prevent seizures [23].

Fasting therapies have been proposed as a treatment for seizure disorders since the time of Hippocrates [23] and were more formally adopted for the treatment of epilepsy in 1911 by the French physicians Guelpa and Marie [24]. In 1921, fasting and low-carbohydrate/high-fat diets were proposed to increase ketone levels in normal, healthy subjects and were recommended to promote therapeutic benefits in epileptic children [25]. A subsequent larger study showed improvement or complete seizure control in 56% of adolescent and adult patients [26].

Currently, ketogenesis finds application in several fields such as the treatment of various neurological conditions both in children [27] and adults [28], gynecology [29], metabolic syndrome (MetS) [29], and is of recent interest in the field of oncology as well [29]. In addition, KD is presently considered the only therapy in the following metabolic disorders: Glucose transporter protein 1 (GLUT-1) deficiency syndrome, Complex 1 mitochondrial disorders (C1MDs), and Pyruvate dehydrogenase deficiency [30].

Ketogenic therapies include any intervention that induces the organism to produce ketones: acetoacetate, β-hydroxybutyrate (BHB), and acetone (Figure 1). These are formed from ingesting fatty acids (while using ketogenic high-fat diets) or producing endogenous fatty acids (as during fasting and in hypoglucidic–hypolipidic–normoprotein diets) [31].

The common goal of all KDs is to induce the production of ketones in the liver through the beta-oxidation of free fatty acids with the aim of mimicking a state of fasting, but without depriving the body of the calories needed to support growth and development [32,33]. The ketones that are generated, acetoacetate and BHB, then enter the bloodstream and are transported to metabolically active tissues, i.e., skeletal muscle, heart, and brain, where they are used for energetic purposes [29], whereas acetone, produced by the spontaneous decarboxylation of acetoacetate, is rapidly eliminated through the lungs and urine (Figure 2).

There are multiple ketogenic dietary interventions are used in clinical practice and evaluated in the management of headaches. Although they have in common the restriction of carbohydrates and adequate protein intake, they differ from each other based on the variation of the lipid to carbohydrate and protein ratio (Figure 3). The various protocols include:1The classic ketogenic diet (CKD), in which the ratio between fats and non-fats (carbohydrates + proteins) must be calculated; generally, this ratio is 3:1 or 4:1 (i.e., the intake in grams of fats is three or four times that of non-fats). This protocol is characterized by the higher content of fats compared to the protein portion (slightly reduced or normal) and carbohydrates (greatly reduced) [27];2The supplementation of medium-chain triglycerides (MCT), in which about 60% of the caloric intake comes from MCT, whose metabolic fate can only be the production of energy; if taken in excess, acetyl-CoA will accumulate, resulting in turn in the biosynthesis of ketones [27];3The modified Atkins diet (MAD), the most liberal in terms of protein intake and the least restrictive in terms of the need to weigh each food [27];4The Very Low-Calorie Ketogenic Diet (VLCKD), an extremely restrictive nutritional protocol (600–800 kcal), limited in time (up to 12 weeks), characterized by a minimum protein content (≥75 g/day), a very limited carbohydrate content (30–50 g/day), a fixed amount of fat (20 g/day, mainly from olive oil and omega-3), and micronutrients to meet the Dietary Reference Intake (DRI), in accordance with the European Food Safety Authority (EFSA) [31,34].

KD protocols are sometimes referred to as Low-Carb High-Fat (LCHF) because of the peculiar distribution of macronutrients. It should be emphasized that KD protocols produce a moderate ketonemia, up to about 5 mM, therefore not comparable to the harmful levels found in ketoacidosis, typically seen in patients with type I Diabetes Mellitus, in which blood ketones reach a range between 10 and 25 mM [35,36].

In addition to the above-mentioned protocols, the Low Glycemic Index Diet (LGIT), characterized by the intake of a higher quantity of carbohydrates (60–80 g/day) distributed over several meals, all coming from low glycemic index sources. This diet, although not strictly ketogenic, has been shown to be effective in some forms of epilepsy [27] and is used in clinical practice in the management of headaches as well.

## 3. Specific Evidence of the Effectiveness of KD in Headaches

### 3.1. Clinical Evidence in Migraine

The first evidence of a potential benefit of the KD diet as a migraine prophylaxis dates back to 1928 [37], starting from the erroneous assumption that migraine was an attenuated form of epilepsy, and that the ketonuria sometimes found at the end of an attack in patients with cyclic vomiting and migraine was a result of a physiological state of ketosis established to lead to the cessation of the attack itself.

A further study in 1930 [38] on 50 migraine patients showed an improvement of headache in 39 patients, with total remission of headache in 50% of treated cases and a reduction of at least 50% of headache in 75% of patients.

The next mention in the literature of the possible benefit of KD in migraine is from 2006 [39] and describes a single clinical case in which a patient with chronic migraine (≥15 days/month) associated with medication overuse headache (MOH) had resolution during a prolonged period of treatment with VLCKD.

Furthermore, two additional cases have been reported [40] of obese patients with episodic migraine (<15 days/month) who improved with a VLCKD. To confirm this observation, the same authors performed a proof-of-concept study [41] comparing two weight-loss diet regimens in 95 patients with episodic migraine: one VLCKD, the other hypocaloric non-ketogenic. The study design followed the group of patients while on KD for a single month, followed by another 5 months of a progressive re-introduction of a non-ketogenic diet: one month of progressive reintroduction of carbohydrates in which patients continued to take nutraceutical supplements (vitamins and minerals) used also during the diet; one month of reintroduction of carbohydrates in the absence of these supplements; 3 months of “standard” weight-loss diet, similar to the one followed by the other group that had not undergone KD. At the end of the study, a slight improvement in migraine parameters was observed in both groups of patients compared to baseline. However, in the group that followed VLCKD, there was a dramatic improvement of all migraine parameters during the ketosis state (as detected by urinary sticks), which promptly regressed once KD was discontinued.

To clarify definitively the direct effect of KD on episodic migraine, a controlled, randomized crossover, double-blind study was conducted comparing 2 isocaloric diets (800 Kcal per day): one VLCKD, the other Very Low-Calorie non-Ketogenic Diet (VLCnKD) in a population of 35 obese migraineurs. At the end of the trial, the response rate (the number of patients with ≥50% reduction in headache frequency) was 74.28% of patients during VLCKD, compared with 5.7% during VLCnKD and this difference was statistically significant [42].

More recently, a group of 14 subjects with episodic migraines underwent the use of a one-month MCT supplementation without any other dietary changes. At the end of the study patients self-reported an improvement in terms of frequency, duration, and symptom severity of migraine [43].

Following this study, the efficacy of a 3-month normo-caloric KD was also tested in a group of 23 patients with MOH [44]. With a baseline frequency of 30 days per month, the frequency decreased to 7.5 days per month; median headache duration reduced from 24 to 5.5 h, and abortive drug consumption from 30 to 6 doses per month.

Few studies with negative results are available examining younger patients with headaches. In particular, it was observed the lack of effect of MAD in adolescents with CDH [45]. This finding was not unexpected because of specific diagnostic and therapeutic challenges present in pediatric patients with headaches. Instead, according to the ICHD-3, the interpretation of pediatric headache, and response to treatment, may not be clear, either because of the child’s difficulty in reliably reporting symptoms or because of their milder manifestations in pediatric migraine [2], therefore, misdiagnosis and misinterpretation are always possible.

Finally, some evidence of LGIT efficacy in patients with migraine were recently reported [46,47].

### 3.2. Clinical Evidence in Cluster Headache

In cluster headache (CH), reports of the efficacy of the ketogenic diet are limited to a single article in the scientific literature [48], although this is a much-discussed topic in various online patient discussion forums [49,50,51,52].

In a case series conducted prospectively for 3 months on 18 patients with drug-resistant chronic cluster headache (CCH) who were asked to follow a MAD, 15 patients were considered responsive to the diet: 11 had complete resolution of headache and 4 had an average reduction in attacks of ≥50% while on the diet. At baseline, the mean monthly number of attacks for each patient was 108.71 (SD = 81.71); at the end of the third month of the diet, it decreased to 31.44 (SD = 84.61) [48]. Although preliminary and not powered to show efficacy, these results seem promising and suggest that CH patients may also benefit from KD.

## 4. Mechanisms of Action of Kd in Treating Headache

Ketogenic diets have been shown to have multiple mechanisms of action, many of which may be relevant in the management of headaches (Figure 4).

### 4.1. Neurophysiologic Modulation

Through the study of evoked potentials (EPs) and brainstem/spinal reflexes, it is possible to measure and differentiate the functioning of distinct brain structures, like the cortex and the trigeminal nucleus caudalis. Through these methods, it is possible to identify and monitor some pathophysiological markers of migraine, including habituation that is a decremental response to repeated non-salient stimuli, common to many biological systems. In most migraine patients, this phenomenon is absent in the intercritical period, whereas it normalizes during an attack. This has been widely demonstrated, both at the cortical and subcortical levels [53].

Thus, by the study of EPs (which analyze cortical activity) and of the blink reflex (which analyzes brainstem activity) it has been observed that regardless of the type of stimulus received (visual, somatosensory and trigeminal nociceptive), during ketogenesis the interictal lack of habituation normalizes in migraine, while the deficit of habituation at the brainstem level persists [54,55]. This observation led the authors to conclude that the action of ketones is exerted at the cortical level, not at the brainstem level, where the so-called “migraine generator” could be localized (from 24 h before the spontaneous migraine attack, hypothalamic activity is increased and altered in functional coupling between the spinal trigeminal nuclei, but with the dorsal rostral pons during the attack) [56]. Therefore, the effect of KD could be exerted downstream of the triggering of the migraine crisis.

### 4.2. Cerebral Energy Metabolism

Several magnetic resonance spectroscopy studies highlight that the brains of migraine subjects are consistently in an energy deficit compared with those of healthy subjects [57,58].

The greater energy efficiency of ketones compared with glucose (100 g glucose generates 8.7 kg ATP, 100 g BHB can produce 10.5 kg ATP, and 100 g acetoacetate 9.4 kg ATP) [59] may alleviate the energy deficit described in the migraine brain.

From a molecular point of view, BHB is metabolized directly in the mitochondria to acetyl-CoA, which in turn is used during oxidative metabolism in the Krebs cycle to produce ATP. At sufficiently high blood concentrations of ketones, under normal glycemic conditions, BHB can meet the entire basal neuronal requirement and approximately half of the activity-dependent neuronal requirement [60]. In general, ketones can produce up to 60–70% of energy requirements during a state of physiological ketosis [61] and are used by synaptic terminals and all brain cells as an energy substrate, especially by neurons and oligodendrocytes, which use them three times more efficiently than astrocytes [62,63]. The availability of higher levels of ATP can improve the above-mentioned migraine-specific brain energetic deficit.

### 4.3. Alteration of Mitochondrial Dysfunction

In addition to improving energy metabolism, KD also improves mitochondrial biogenesis, as has been documented in animal model studies. Specifically, KD has been shown to promote the conversion of adipose tissue to brown fat, with an increase in median adipose tissue size and cAMP values of approximately 60%, a 2.5-fold increase in cAMP-binding proteins (suggestive of increased sympathetic activity that would cause increased lipolysis), and an increase in mitochondrial oxidative phosphorylation proteins [64]. Moreover, KD has been shown to restore mitochondrial Complex 1 function in C1MDs [65]. All of this would result in increased mitochondrial size and increased efficiency of lipolytic mechanisms. At the same time, this is associated with a halving of plasma levels of insulin and leptin [64], which by inducing insulin resistance contributes to the genesis of headache [66].

Furthermore, KD improves mitochondrial membrane permeability. This allows for greater exchanges, facilitating the supply of energy substrate and the release of waste and oxidative products that could damage mitochondria [67,68].

Mitochondrial dysfunctions have also been shown in patients with migraines, and recovery of function correlated with an improvement in migraine symptoms [69,70]. In particular, migraine improves with high doses of riboflavin that act in the recovery of deficiencies of the electron transport chain of mitochondrial Complex 1 [69] that also is a selective metabolic target of KD. The effect of ketones as mitochondrial boosted could account for their protective effect in migraines.

### 4.4. Oxidative Stress

Migraine patients (in both chronic and episodic forms) exhibit increased oxidative stress [65,67,71] and oxidative stress has been hypothesized to be the common point among all the mechanisms of action by which the different migraine triggers induce the initiation of the migraine attack [72]. Indeed, the use of antioxidant nutraceutical compounds has been widely adopted by migraine patients [73] and exogenous ketones are among these products. In fact, BHB reduces the production of reactive oxygen species [74] through its action on mitochondrial complex II [75,76]; furthermore, due to the increased heat of combustion of BHB relative to that of pyruvate, BHB also increases the efficiency of ATP production from the mitochondrial proton gradient and reduces free radical production [77]. BHB has also been shown to reduce lipoperoxidation caused by three days of intrastriatal glutamate injections in mice [78]. Moreover, the synthesis of 2 molecules of acetyl-CoA to be used for energy purposes by glucose involves the conversion of 4 molecules of NAD+ to NADH: 2 molecules of NAD+ are reduced at the cytoplasmic level, 2 at the mitochondrial level. On the contrary, the biosynthesis of the same 2 molecules of acetyl-CoA from BHB involves only the reduction of 2 molecules of NAD+ to NADH, both in the mitochondrion, preserving the cytoplasmic NAD+ pool (7) and thus preventing cellular aging by an epigenetic mechanism (inhibition of histone deacetylase) [79].

### 4.5. Anti-Inflammatory Mechanisms

Inflammatory phenomena related to nitric oxide (NO) pathways and involving nuclear factor-kappaB (NF-KB) are among the central drivers in migraine pathophysiology [80] by the activation of that transcription factor in the nucleus trigeminalis caudalis [81]. This inflammatory process is inhibited by the agonism of the hydroxy-carboxylic acid receptor 2 (HCA2) [82] that is, among other things, expressed in dendritic cells and neuroglia, and has as endogenous ligand the BHB [83,84]. Hence, BHB could lead to an inhibition of neuroinflammatory phenomena at the onset of migraine attacks by the activation of HCA2 receptors that in turn reduces the proinflammatory stimulation of NF-KB induced by NO signaling.

Moreover, BHB in immune cells has an inhibitory effect on the inflammosome, a protein complex involved in the intracellular genesis of inflammation, reducing the production of inflammatory cytokines and consequently inflammation [85]. it is possible to hypothesize that also the inflammosome is involved in the inflammatory mechanisms typical of migraine because it has been related to headache manifestations having the characteristics of migraine [86,87].

### 4.6. Epigenetics

Histone hyperacetylation is generally associated with activation of gene expression; therefore, class I histone deacetylase (HDAC) activity suppresses such expression. BHB is structurally similar to butyrate, the canonical HDAC inhibitor [88]. Fasting, which increases plasma levels of BHB, is associated with increased histone acetylation in a number of tissues [79], including nerve tissue [89]. BHB has been shown to be an inhibitor of HDAC (HDAC1, HDAC3, and HDAC4 class I and II types) by a dose-dependent mechanism, resulting in upregulation of genes involved in the FOXO3A network, including catalase, mitochondrial superoxide dismutase (Mn-SOD), and metallothionein 2. This results in a gene expression mediated protection against oxidative stress and by regulating metabolism [79], both mechanisms involved in migraine pathophysiology, as discussed above.

BHB also regulates BDNF (brain-derived neurotrophic factor) expression in the brain, particularly in the hippocampus. In addition to KD, exercise also increases BHB levels and BDNF expression in the hippocampus [89]. BDNF was related to migraine susceptibility both in episodic [90] and chronic forms [91], potentially by BDNF-induced pain-related neural plasticity [92], and its modulation by BHB can improve clinical symptoms.

Another mechanism of epigenetic control is exerted by the so-called micro-RNA (miRNA). It was recently observed that KD can modulate miRNAs through the promotion of antioxidant and anti-inflammatory biochemical pathways [93] that in turn lead to protection from migraines.

Finally, DNA methylation, the most known epigenetic mechanism, has been shown to be involved in the regulation of migraine pathophysiology by the modulation of gene expression of CGRP [94]. KD is able to modify the methylation state of genes and this mechanism of action is proposed to reduce seizures [95]. A similar mechanism of action may also reduce migraine frequency and severity.

### 4.7. Cortical Spreading Depression (CSD)

KD also has a protective effect on CSD because ketones reduce the propagation rate of CSD [96,97], as well as making its initiation less likely, by correcting the energy deficit (see below).

CSD is well-established in the pathophysiology of migraine (at least in migraine with aura).

The gap between low ATP levels and excessive neuronal activation during the interictal phase [98,99] could lead to a metabolic imbalance capable of promoting CSD and activating the trigeminovascular system, thus triggering a migraine attack [100,101,102].

### 4.8. Glutamate/GABA Balance

BHB and KD both induce increased biosynthesis of gamma-amino hydroxybutyric acid (GABA, the most important inhibitory neurotransmitter) via a decarboxylation process that irreversibly converts glutamate (GLU). In addition, further depletion of the brain concentration of GLU may be due to its use for energy purposes when individuals are on KD [97] and reduction in intracerebral GLU concentrations. This is important because GLU in addition to being an amino acid is an excitatory neurotransmitter, known to be a trigger for migraine [103,104,105]. Moreover, the cerebral cortex of migraine patients has a higher baseline concentration of GLU [106]. Because of the above, it can hypothesized that KD reduces both absolute GLU concentration and glutamatergic excitatory activity due to the increased concentration of GABA, which cortical inhibition, resulting in a potential protective effect on preventing migraine.

### 4.9. Ion Channels

One of the mechanisms of action proposed to explain the effect of ketogenic diet concerns the modulation of ion channels, in particular the adenosine triphosphate-sensitive potassium channels (K_ATP_ channels) that are opened by KD metabolites, reducing firing in central neurons [107]. K_ATP_ channels are being evaluated with interest as possible novel therapeutic targets for migraine because of their involvement in migraine pathophysiology [108]. On the other hand, KD has shown to be effective in alternating hemiplegia of childhood related to ATPase Na+/K+ Transporting Subunit Alpha 3 (ATP1A3) gene mutations. ATP1A3 gene mutations alter the ionic currents across the cell membrane and account for a wide spectrum of neurological disorders, including hemiplegic migraine [109].

### 4.10. Gut–Brain Axis

Migraine can be triggered or exacerbated by gastrointestinal disorders (nausea, dyspepsia). Gastric dysfunction may promote alterations of the microbiota with the development of putrefactive processes and consequent local inflammatory reaction [110,111]. This phlogistic state could induce the activation of vagal afferents towards the hypothalamus [112,113] leading to a worsening of migraine [114]. In support of this hypothesis, vagal inhibitory modulation has a beneficial effect on migraine [115,116,117,118], cluster headache [119], and seizure reduction [120].

KD, besides resulting in sympathomimetic activity [121] (thus counterbalancing vagal parasympathetic hyperactivity), also acts on the regulation of the intestinal microbiota [122]. Two double-blind studies on the efficacy of probiotic supplementation in migraine [123,124], shed a light on the possible role of gut microbiota in the pathogenesis of this form of headache, maybe mediating KD efficacy in migraine, as proposed for epilepsy [125,126,127].

### 4.11. Intracerebral Glucose Metabolism

A further possible mechanism of action involved in the efficacy of KD in migraine is related to a potential improvement of patients’ intracerebral glucose metabolism. In particular, we speculate about a role for the SLC2A1 gene, accounting for GLUT-1 deficiency syndrome, an autosomal recessive disorder in which the type 1 glucose transporter (responsible for crossing the blood–brain barrier of this sugar) has reduced or absent function. Several cases have been reported in the literature in which migraine was part of the symptom spectrum of GLUT-1 deficiency syndrome [128] and regressed following the establishment of a KD. It could be supposed that the presence in heterozygosity of polymorphisms or mutations in this gene may lead to intermediate phenotypes due to decreased sugar supply to the brain, leading to the development of a clinical parade of neuropsychiatric symptoms, including migraine. The only existing therapy in GLUT-1 deficiency syndrome is KD, because ketones do not require the activity of this transporter to cross the blood–brain barrier, bypassing the metabolic blockade and restoring proper energy metabolism [129,130]. Like this pathology, it is possible to hypothesize a protective role of KD from any abnormality of glucose metabolism in migraine subjects.

### 4.12. Migraine and Metabolic Syndrome

MetS is characterized by criteria defined in a consensus document published in 2009 by the IDF, NHLBI, AHA, WHF, and IASO [131].

Harmonized criteria for diagnosis:-obesity, defined as abdominal circumference (≥95 cm man and ≥80 cm woman) plus at least 2 of the criteria below:-hypertension (blood pressure ≥130/85 mmHg or patient on antihypertensive therapy);-fasting blood glucose (≥110 mg/dL);-triglyceridemia (≥150 mg/dL);-low plasma HDL levels (<40 mg/dL man and <50 mg/dL woman).

Individuals with MetS are at higher risk of developing cerebrovascular and cardiovascular diseases, diabetes, cancer, polycystic ovary syndrome, etc.

In addition, MetS is more prevalent in migraineurs (21.8% with aura, 16.8% without aura) than in the general population (14.5%), [132] and is related to the development of its chronicity, especially of MOH [133]. Among the characteristics of MetS, those most correlated with migraine with and without aura are low HDL cholesterol levels, hyperglycemia, and excess visceral fat [134].

Migraineurs exhibit a tendency toward hyperinsulinism compared with healthy subjects and those with other forms of headache [135,136], and they have more cerebrocardiovascular events (which is the expected outcome of MetS) and more risk factors for such events [137].

Obese and underweight individuals are both at increased risk of developing chronic [138]. In general, there is a correlation between the frequency of migraine attacks and two weight measures, BMI and waist circumference; this association is stronger in patients on prophylactic therapy, who are also more frequently found to be overweight [139]. In contrast, in obese subjects a high total fat free mass (lean mass) would seem to be a protective factor against the development of migraine [140]. In addition, it has been reported that weight loss may lead to an improvement in the frequency of migraine attacks [41,141,142]. Therefore, it is conceivable that reducing fat mass and preserving/increasing lean mass may be protective against migraine. This has been achieved with VLCKD-type protocols [143].

Interestingly, weight gain is a common side effect of most migraine prophylaxis treatments. In particular, flunarizine, valproic acid, and amitriptyline induce weight gain related to higher levels of insulin, leptin, and peptide C [144], along with changes in hypothalamic orexinergic peptide levels [145,146,147]. The above-mentioned changes induced by the use of preventive treatments for migraine, in particular weight gain [138], alterations of insulin and leptin levels [66], and the development of leptin resistance (which in turn is potentially responsible for worsening headaches regardless of weight gain [148]), could counteract the efficacy of therapies, leading in the long term to a worsening of the pre-existing migraine (in a sort of “prophylactic paradox”) and therefore to the discontinuation of therapy for ineffectiveness and/or weight gain. In fact, it has been reported that patients on preventive therapy for migraine tend to develop a worsening of metabolic parameters which correlate with the worsening of headache [139].

## 5. Study Group Recommendations on the Management of Headache Patients Using a Ketogenic Diet

A group of experts in the field of headache with a specific interest in KD was identified by Drs. Cherubino Di Lorenzo and Giulio Sirianni, based on contributions to scientific publications, congress proceedings, and direct knowledge of the prescription of KD to treat headache. Through this selection, specialists from eight Headache Centers were identified and agreed to participate in a working group (Azienda Ospedaliero-Universitaria Consorziale Policlinico di Bari, IRCCS—Ospedale Bellaria Carlo Alberto Pizzardi of Bologna, Piero Palagi Hospital of Florence, Headache Center of the Polo Pontino of the Sapienza University of Rome, San Carlo Borromeo Hospital of Milan, IRCCS—San Raffaele Pisana of Rome, Humanitas Gradenigo Hospital of Turin, Azienda Ospedaliera Santa Maria della Misericordia of Udine). All centers have been treating headache patients with KD for a period of one year or more (range from one to ten years). Four Centers have a dedicated Keto-Team, or in-house medical consultants and/or dietitians experienced in the use of KD.

Participants were asked to complete a questionnaire about their direct experience with the specific topic. Responses and comments were collected by Dr. Grazia Semeraro and incorporated into a document that was debated in two roundtables discussions by the group of experts. Upon drafting of this work. The clinical recommendations were revised by 3 external experts, one in the field of migraine (Gianluca Coppola), another one in the field of adult neurological indications for KD therapies (Mackenzie C Cervenka), and the last in the field of MetS (Giovanni Spera), to produce the final manuscript.

## 6. Recommendations

Based on up-to-date data, it is not possible to assess all of the possible applications of KD in headache disorders. Nevertheless, the above-mentioned evidence suggests that it may be beneficial in a variety of headache disorders.

### 6.1. Patient Selection

The expert panel agreed that patients with migraine appropriate for referral for KD, as complementary and supportive to other NPAs, includes those with and without aura, both in the episodic form [(≥4 days/month, <15 days/month), analogous to other prophylaxis as reported by SISC and ANIRCEF (the 2 Italian scientific societies dealing with headache) guidelines [149,150] and patients with chronic (≥15 days/month), MOH and those with CH, episodic in the active phase or chronic. Within these diagnostic groups, those who are overweight, obese or with MetS, and patients with ineffectiveness, poor tolerability, or contraindications (evidenced by history or medical records) to prophylactic drug therapies are considered KD candidates. In addition, patients who have expressly asked the specialist to start this diet therapy, not wanting to undergo pharmacological prophylaxis. In addition, KD could also be reconsidered in cases in which a previous dietary failure was attributable to the inadequacy of the dietary protocol, which was not tailored to the individual or the therapeutic indication for headache (see below). Although none of the board members has extensive experience treating children and adolescents with diet therapies for headache, the large body of scientific literature available on the efficacy of KD in neurological disorders in children and reviewed in this manuscript support potential application in early life. However, more studies evaluating safety, feasibility, and efficacy of ketogenic diet therapies in the management of headaches in children and adolescents are needed [151].

In the selection phase, patients should be made aware of the goals to be achieved: reduction of headache days by at least 50%, reduction of analgesic consumption and of any prophylactic drugs. The patient must be informed and fully aware of the commitment he/she must make in following the diet therapy.

### 6.2. Multidisciplinary Evaluation and Diet Therapy

All participants in the working group agreed that KD initiation should first involve clinical evaluation by a headache specialist, followed by screening and approval by the nutrition professional (Physician Nutrition Specialist and Registered Dietician) who would develop and individualize the diet.

Hence, in clinical practice in all the headache centers contributing to the board, once the patient has been selected for evaluation by the nutrition professional, the diet is started only after an outpatient visit by the latter to confirm suitability for KD, and to elaborate and explain the diet therapy protocol. The choice of the type of diet to be followed by the patient is also determined according to the patient’s preferences, in order to increase his/her ability to adhere to the new dietary plan, so as not to create a reduction in the quality of home and work life. The board agreed that it is not necessary to hospitalize the patient or undergo a preventive fast at the time of diet initiation.

The success of diet therapy is often dependent on initiation practice. Different approaches have been proposed, from the sudden change of the diet to a more gradual start with an LGIT or MAD with an initial ratio of about 1:1, and then increasing the fat and reducing the glucose and protein intake. In cases where there is a transient worsening of clinical symptoms in the first days of the diet, additional meals can be provided, in addition to any pharmacological therapy to control headaches. The board concluded that there is insufficient evidence to guide the selection of the best type of diet to attempt first, both because each group utilized different diets or diet combination, and because the decision was left primarily to the nutrition professional, considering the preferences of the individual patient. If there is no response to the diet within 3 months, the ketogenic ratio can be increased further if tolerated. If ineffectiveness persists, the diet is typically discontinued after month 6.

KD requires specific integrations of minerals (cations) and vitamins because intake does not otherwise meet recommended quantities on carbohydrate-restricted diets. Patients with headaches may present with comorbidities such as other neurological disorders that may benefit from or be exacerbated by KD. For instance, magnesium and folic acid deficiencies, may be a cause of, or associated with a worsening of headache [152]. In KD, especially for LCHF diets, Omega-3 supplementation may be useful to improve the lipid profile and possibly have an additive protective effect on headache (a finding proposed by several authors but not yet confirmed) [153].

By a review of each center’s practice, it was concluded that the types of diet used in the treatment of headache were similar to those used in the field of epilepsy, with some distinctions.

1Unlike other neurological diseases, migraine can in some cases be triggered by specific foods; in general, excessive use of foods containing biogenic amines (aged cheeses and sausages), especially histamine (nuts), monosodium glutamate and processed foods should be avoided [154]. In addition, some patients have reported worsening headaches when consuming foods with gluten additives, excess fermentable oligo- and mono-saccharides and polyols (FODMAP) or using seed oil.2In Italy, the VLCKD protocols are widely used for weight loss, generally not used in the neurological field as ketogenic therapies. However, these protocols have been used on numerous obese patients by all centers in our working group and their efficacy on migraine has been repeatedly reported in the literature [39,40,41,42]. The use of these diets should be limited to no more than 12 consecutive weeks [34], at the end of which the patient either exits the state of ketosis (even receiving the indication to follow a maintenance diet of LGIT or Mediterranean type without added sugars), or transit to a normo-caloric KD of longer duration.3In the field of epilepsy, the use of 3:1 CKD is commonly used in children, as can be seen in the literature. On the contrary, since the centers involved in this board are mainly active in the treatment of adult subjects, few of their patients have been prescribed a CKD with a high ketogenic ratio (≥3:1) because of the difficulty of matching the caloric requirement to protein needs (for instance, in case of a protein need of 70 gr and considering a daily intake of 30 gr of carbohydrates, to fulfill the 3:1 ratio should include 300 gr of fats, for a total daily intake of 3100 kcal). Therefore, the most used diets were MKD (maintaining a ketogenic ratio around 2:1), MCTD, and VLCKD (for obese patients).4Although exogenous ketone sources (salts or esters) are already commercially available and regarded as a promising therapy in both epilepsy [155] and migraine [21], no center of those involved in the working group has clinical experience with the use of these supplements. While waiting for data from clinical trials [156], some concerns were raised however about their use as it may only confer a partial effect compared to that of a KD. In fact, the therapeutic action of the diet is not only due to the role played by ketones, but also to the change in the macronutrients consumed (see LGIT diet) and the correction of insulin resistance typical of migraine sufferers. Therefore, the board expresses some skepticism and concluded that further investigations is warranted.

An overview of recommendations is summarized in Table 1.

### 6.3. Contraindications

Absolute and relative contraindications to treatment should be ruled out before having patients start a KD (Table 2) [30]. If there is a contraindication to KD, an LGIT may be a reasonable alternative. Contraindications may be discovered after starting the diet, as is the case of pregnancy. In the absence of data on the safety of a KD approach for the mother and fetus [157], when pregnancy occurred during KD, the patient should be immediately transitioned to an LGIT diet, but with an adequate amount of carbohydrates to avoid ketosis.

Socioeconomic and family support networks should also be taken into account before proposing the diet to the patient as lack of support may be considered a relative contraindication.

### 6.4. Patient Monitoring

Before starting the diet, the patient should undergo an electrocardiogram and a complex laboratory evaluation (Table 3). These evaluations should be repeated every six months, while in the case of diets of longer duration (>12 months) the board proposes to also perform a bone mineral density scan and Bone Densitometry and abdominal ultrasound.

It is also recommended that nutrition professionals obtain anthropometric measurements (including waist circumference and plicometry) and possibly perform an examination of body composition (e.g., Bioimpedance).

The working group agreed that patients undergoing KD should keep a headache diary during treatment to monitor headache parameters and a food diary documenting food ingested and monitor body weight. Ketone monitoring (blood BHB or urine AA) is recommended during the early months of KD as an objective indication of KD compliance and biochemical response as reported in the International Recommendations for the Management of Adults Treated with Ketogenic Diet Therapies [28]. The working group also recommended, in case of lack or loss of response to dietary therapy, to measure the production of ketones, mainly through urine (cheapest and most common method), but also through the exhaled or capillary blood. It is recommended that blood glucose be re-evaluated when needed in case of weakness, dizziness, sweating, or other physical symptoms suggestive of hypoglycemia.

### 6.5. Side Effects

In general, the centers involved in the board rarely found side effects in patients who closely followed their assigned diet. Among the most frequent are muscle cramps, weakness, hypotension, constipation, and unintended weight loss. To correct the first four, it is almost always sufficient to adjust the supplementation of specific minerals and increase hydration. In the case of unintended weight loss, if it is not possible to correct it by increase calories through increased fat intake, it may be necessary for the patient to abandon the diet, but the patient can be directed towards an LGIT diet. The side effects rarely encountered were hyperlipidemia (generally transitory and well-controlled by the consumption of Omega-3 or lipid-lowering drugs), gallstones (especially in case of patients with excessive weight loss, treatable with ursodeoxycholic acid), menstrual irregularity (mainly in VLCKD protocols), as well as alopecia and nail fragility (usually after long periods of diet, treatable with specific supplements). Gastrointestinal symptoms, such as nausea, vomiting, abdominal pain and diarrhea were more rarely reported. On the other hand, while reported in the literature, neither prurigo pigmentosa (the so-called keto rash) [158], nor mood alterations [159] due to the prolonged duration of the weight loss diet were observed by members of the bord in their practice (Table 4).

### 6.6. Causes for Reduced Adherence, Compliance and Diet Cessation

The reasons that most frequently lead patients to reduce compliance with the diet or decide to stop were found to be, in order: inadequate response to treatment, intolerance to carbohydrate restriction or craving for carbohydrates, deviating too much from dietary preferences, monotony and poor dietary variability, disruption of family rituals and habits (eating the same meal together/having to cook different dishes), the creation of problems at work (embarrassment in eating differently from colleagues/inability to follow strict respect meal times or protocols), cost, difficulty in managing a social life and continuing the diet during vacations, lack of a dedicated care-giver, time required to shop and prepare meals, the persistence of some residual attacks despite the diet.

When present, these issues should be explored and addressed with the patient to improve adherence to the diet. Inadequate response to treatment may depend on characteristics and challenges specific to the patient with headache, which the nutrition professional can identify and address. Several centers reported that KD was effective only after food triggers were eliminated in some. The nutrition specialist should help the patient with finding raw materials, explaining which foods to choose to avoid triggers, and providing a variety of recipes or substitutes to satisfy preferences. With adequate patient training, it is possible to teach him/her what to eat when he/she is away from home and in social contexts, thus eliminating these factors that hinder the success of the diet (holidays/social events). In fact, the time needed to dedicate to the diet is almost always not significantly increased with adequate training. Furthermore, the existence of ready-to-eat meal replacement products on the market could even save time and make it more practical to follow the diet, even for those who do not have the support of a dedicated caregiver. The last element that could contribute to reducing the patient’s compliance is the high cost. Certainly, the lack of a dedicated keto team and therefore the need to recruit providers individually can greatly increase the costs for the patient. On the other hand, even prophylactic drug therapies require monitoring, an aspect that is often underestimated by headache specialists. Concerning the direct cost of the diet, a diet based on good quality food (possibly organic) is comparable to the cost of a KD.

### 6.7. Duration of the Diet

Based on the clinical experience of the board members, patients’ response to KD is typically very quick, generally within one week. However, sometimes improvement is more delayed. If there is no response to the diet within 3 months, increasing the ketogenic ratio could be considered. In patients with a lack of response in the presence of MOH, the continuation of the diet while discontinuing the drug responsible for MOH without corticosteroids may be effective. If lack of efficacy persists, the diet can be discontinued by month 6. There is no optimal duration for the diet that can be generalized to all patients with headaches. In case of efficacy, the minimum duration of the diet should be at least 3 months (as for all prophylactic therapies), so when the target weight is reached in overweight/obese subjects, the caloric intake should be modified to allow for the continuation of the KD, if starting with a weight-loss diet. In patients with episodic CH in the active phase, the diet can be suspended at least one month after the presumed end of the cluster, anyway not before one month after the last attack. In episodic and/or chronic migraineurs and in those with chronic CH resistant to other prophylactic therapies, the duration of treatment should be determined with the patient according to their preferences. The experience of Headache Centers suggests that at the end of therapy there may be a transient persistence of benefit directly proportional to the quantity of weight lost (in the case of obese patients, for diets of shorter duration), to the duration of the diet (in the case of normal weight subjects), and to the effectiveness of KD as a detoxification treatment in MOH. The group agreed that after three months of KD followed by stopping, continued benefit tends to occur in about 20% of patients, after six months in about 30% of patients (in both cases these benefits persist more in subjects who have lost a lot of kg), after 12 months in about 50% of patients (regardless of weight loss). For longer durations of KD no clearly greater efficacy persistence has been observed and in general, after one/two years from stopping the diet, the headache tends to worsen again. Therefore, it is advisable to invite patients to discontinue treatment after 12 months in order to take advantage of the possible transitory persistence of the benefit even by suspending treatment. If desired, KD can be cycled just like other prophylactic headache therapies. In case of headache recurrence upon discontinuation, especially for patients with chronic headaches, the diet can be restarted. To date, there are some patients who have been on a KD for headache continuously for more than six consecutive years without discontinuation.

There is no unique protocol for tapering or discontinuing the diet. In the case of LCHF protocols, fats must be progressively reduced; in the case of VLCKD, fats must be increased. In both cases, there is a tendency to reach a 1:1 ketogenic ratio and then transition to an LGIT for a variable period of time to allow the progressive increase in carbohydrate. Although most patients were advised to ultimately follow a Mediterranean-type maintenance diet, emphasizing low-glycemic index carbohydrate sources, many patients subsequently returned to a regular diet, sometimes re-adopting bad eating habits, although others preferred to continue to follow an “Atkins” or “Paleo” style diet or a “Zone Diet”.

### 6.8. Pharmacological Management

KD should be seen as an adjunctive therapy to any existing pharmacological, which could be subsequently reduced if headache improved on KD. In fact, there are no absolute contraindications to the use of symptomatic and prophylactic headache medications during KD protocols, but attention must always be paid to the carbohydrate content of the pharmaceuticals, especially in liquid and sachet formulations; intramuscular vials and suppositories do not contain carbohydrates, while tablets have small amounts.

Although the coadministration of a KD with common pharmacologic therapies is generally judged to be safe, some additional considerations should be kept in mind:1Topiramate may interfere with renal function by promoting nephrolithiasis and modifying the proper excretion of ketones through the urine, facilitating metabolic acidosis. In addition, it can cause weakness and electrolyte abnormalities related to diet. Its use should be carefully monitored, possibly limited and, where feasible, the dosage should be scaled back in subjects who were already taking it before starting a KD.2Valproic acid can alter hepatic metabolism, potentially limiting ketone bodies’ production. The use of this drug should also be monitored or limited during KD.3Beta-blockers and verapamil may result in bradycardia and weakness that could be exacerbated by KD, in which case drug dosages should be reduced.4Corticosteroids, often used as salvage therapy in status migrainosus (a debilitating migraine attack lasting for more than 72 h), as a strategy to interrupt the medication overuse in MOH, and as prevention in CH, may interfere with ketogenesis because of its impact on hepatic functioning and produce hyperglycemia. Therefore, the use of steroids should be carefully evaluated, severely limited, and monitored during a KD.5Flunarizine and amitriptyline, in addition to causing ECG alterations, may, along with valproic acid, result in increased appetite and weight. This could impede compliance with the diet and with progress in losing weight. These aspects should be considered in the case of co-administration.6Although symptomatic drugs (non-steroidal anti-inflammatory drugs (NSAIDs), Triptans, Ergot-derivatives, combination drugs) are not contraindicated in KD, sometimes their overuse can nullify the diet’s preventive effect on migraine. From the experience of centers that have treated many patients with MOH using KD, even when patients do not halt medication overuse, it appears that in more than 50% of cases the diet is effective in blocking overuse of analgesics and headache chronicity. If this does not happen, a medication overuse withdrawal without corticosteroids can be combined without interrupting the diet, often with excellent results even in patients in whom previous attempts at medication overuse end have failed. In general, however, caution should be exercised with the overuse of NSAIDs and acetaminophen because of the impact these drugs may have on renal and hepatic metabolism.

### 6.9. Future Perspectives

Possible future developments emerged from the discussion of the working group on KD use in patients with headaches. First, there is a lack of supporting evidence compared to the study in epilepsy, for which there are many publications and randomized clinical trials. Therefore, it is envisioned that this type of scientific research development can also be carried out in the field of headache, in order to accumulate more supporting evidence on the topic. In particular, it would be interesting to evaluate the impact of an MKD in migraine and CH in double-blind studies. We anticipate that future revisions of these recommendations will be based not on expert opinion but on evidence from rigorous clinical trials.

An additional topic to be studied in these KD responsive patients with a headache would be the effect of supplementation of MCT supplements with a more palatable and liberal diet, such as LGIT, to see if there could be a benefit for reducing the frequency or severity of such headaches as well with this approach. Still to be evaluated is the role of KD in other rarer forms of trigeminal autonomic headaches, such as paroxysmal hemicrania and continuous hemicrania. Similarly, the effect of KD in tension-type headaches should be explored: the working group agreed that this form of headache does not typically benefit from this approach, but there is a lack of studies ratifying its ineffectiveness.

Another challenge identified by the working group in providing patients with access to a ketogenic diet was the difficulty for a Headache Center to establish a Keto-Team, which is essential in the dietary management of patients. In particular, a Physician Nutrition Specialist and Registered Dietician in addition to any other medical professional figures such as an internist, endocrinologist, dedicated nurse and psychologist, are necessary for the creation of the Keto-Team.

To facilitate patient access to this type of care, one consideration is to create specific complex care pathways to automate, after the recommendation of the headache specialist to begin dietary therapy, the initiation and maintenance of KD. It could be the task of a dedicated nurse or other healthcare providers to coordinate the schedule of appointments with the various professionals involved: the physician who assesses the patient’s suitability for the diet, periodic meetings with the nutrition professional, and follow-up visits with the headache specialist. The implementation of these pathways within public or affiliated healthcare facilities could, in addition to simplifying the process for the patient, reduce associated costs.

To increase utilization of these diet therapies and improve patient compliance and adherence, it would be appropriate to involve headache patients’ associations and organize with their courses of ketogenic cooking and information material, both printed and digital. The group of experts hopes that in the future it will be possible to organize specific training materials for patients and their families within the centers and provide them with standardized materials that guide their use of the KD.

## Figures and Tables

**Figure 1 nutrients-13-02307-f001:**
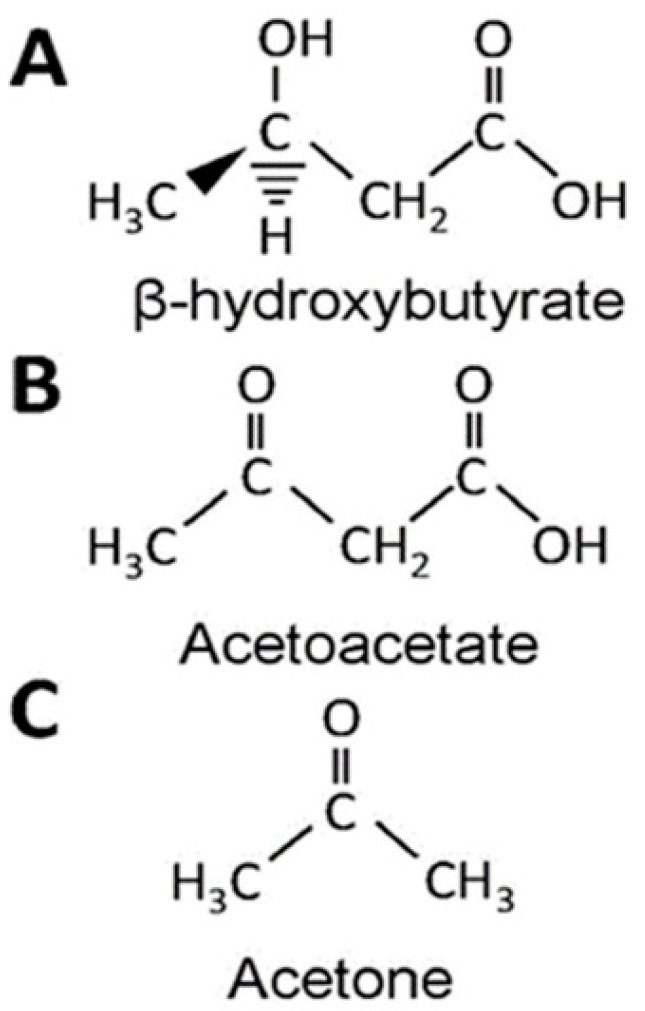
Ketones. A = β-hydroxybutyrate; B = Acetoacetate; C = Acetone.

**Figure 2 nutrients-13-02307-f002:**
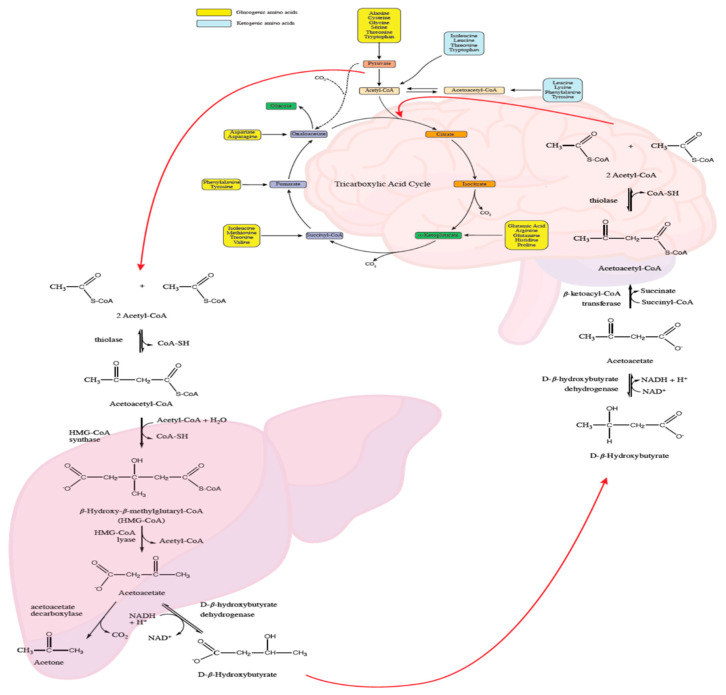
Biochemistry of intrahepatic ketogenesis and energy metabolism of ketones. CoA-SH = Coenzyme A; NAD^+^ = Nicotinamide adenine dinucleotide (oxidized form); NADH = Nicotinamide adenine dinucleotide (reduced form).

**Figure 3 nutrients-13-02307-f003:**
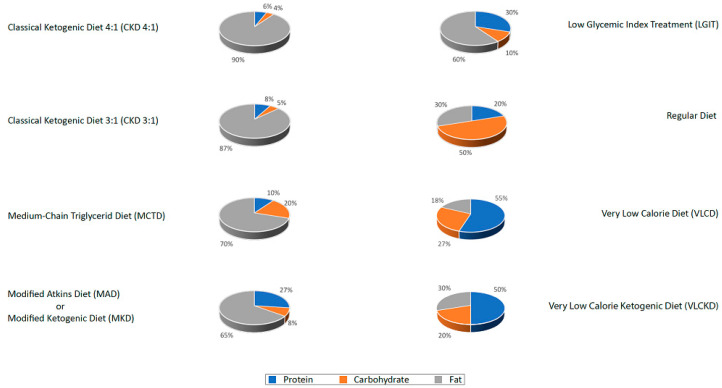
Macronutrient composition of dietary interventions.

**Figure 4 nutrients-13-02307-f004:**
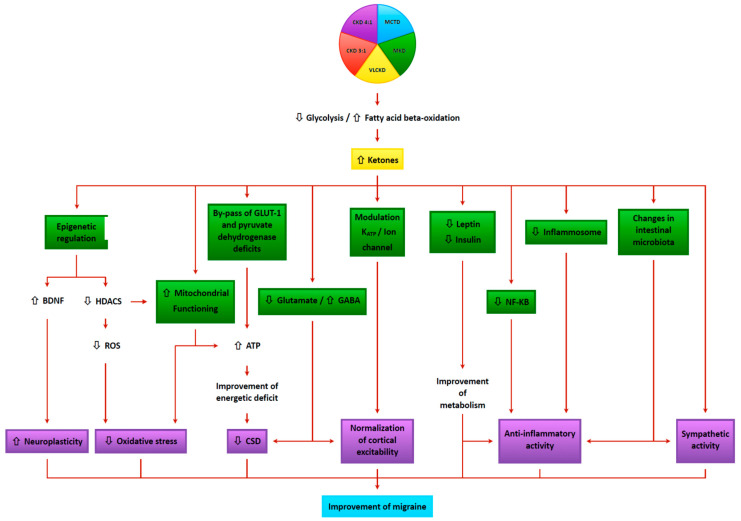
Potential mechanisms of action of the ketogenic diets in the management of headache. ATP = Adenosine triphosphate; BDNF = Brain Derived Neurotrophic Factor; CKD 3:1 = Classic ketogenic diet with ketogenic ratio 3:1; CKD 4:1 = Classic ketogenic diet with ketogenic ratio 4: 1; CSD = Cortical Spreading Depression; GABA = Gamma-aminobutyric acid; GLUT-1 = Glucose transporter protein 1; HDAC = Histone deacetylase; K_ATP_ = adenosine triphosphate-sensitive potassium channels; LGIT = Low glycemic index diet; MCTD = Medium chain triglyceride diet; MKD = Modified ketogenic diet; ROS = Reactive oxygen species; VLCKD = Very low-calorie ketogenic diet.

**Table 1 nutrients-13-02307-t001:** Overview of recommendations.

DIAGNOSIS
Migraine
MOH
CH
**INDICATIONS**
Overweight/Obese
Metabolic Syndrome
Drug resistance/non-tolerability
Patients’ specific request
**AGE**
Adulthood
Lack of data in childhood
**TYPE OF DIET**
CKD 4:1
CKD 3:1
MKD
MCTD
LGIT
VLCKD (only in case of overweight or obesity)

MOH = Medication Overuse Headache; CH = Cluster Headache; CKD 4:1 = Classic Ketogenic Diet with a 4:1 ratio; CKD 3:1 = Classic Ketogenic Diet with a 3:1 ratio; MKD = Modified Ketogenic Diet; MCTD = Medium-Chain Triglycerides Diet; LGIT = Low Glycemic Index Diet; VLCKD = Very Low-Calorie Ketogenic Diet.

**Table 2 nutrients-13-02307-t002:** Absolute and relative contraindications to ketogenic diet.

ABSOLUTE CONTRAINDICATIONS	RELATIVE CONTRAINDICATIONS
Carnitine deficiency (primary) Carnitine palmitoyltransferase (CPT) I or II deficiency Carnitine translocase deficiency β-oxidation dfects Medium-chain acyl dehydrogenase deficiency Long-chain acyl dehydrogenase deficiency Short-chain acyl dehydrogenase deficiency Long-chain 3-hydroxyacyl-CoA deficiency Medium-chain 3-hydroxyacyl-CoA deficiency Pyruvate carboxylase deficiency Porphyria	Pregnancy and breastfeeding Renal failure Severe nephrolithiasis Hepatic failure Pancreatitis Type1 Diabetes Mellitus Arrhythmias Angina Recent myocardial infarction Severe osteoporosis Alcoholism Eating disorder Poor compliance Inability to maintain adequate nutrition

**Table 3 nutrients-13-02307-t003:** Patient monitoring.

MONITORING				
Before Starting	Every Day	Every 6 Months	Every 12 Months	In Case of Non-Response to KD
**Blood Laboratory assessment** Complete blood count Glucose * Basal insulin OGTT (Glucose level, Insulin level) Total cholesterol LDL, HDL, Tg Direct bilirubin Indirect bilirubin GOT e GPT Uric Acid/Nitrogen Creatinine Homocysteine Protein electrophoresis Na, K, Cl, Ca, P, Mg Folates Vitamin B12 Vitamin D Urinalysis **EKG** **Blood pressure** **Weight**	**Headache diaries** **Eating diaries**	**Blood Laboratory assessment** Complete blood count Glucose * Basal insulin Total cholesterol LDL, HDL, Tg Direct bilirubin Indirect bilirubin GOT e GPT Uric Acid/Nitrogen Creatinine Homocysteine Protein electrophoresis Na, K, Cl, Ca, P, Mg Folates Vitamin B12 Vitamin D Urinalysis **EKG** **Blood pressure** **Weight**	**BMD/DEXA** **Abdominal ultrasonography**	**Evaluation of ketones (blood, urine or breath)**

* Extemporaneous assessment in case of asthenia, dizziness, sweating, or other physical symptoms. LDL = Low-density lipoprotein; HDL = High-density lipoprotein; Tg = Triglycerides; GOT = Glutamic-oxalacetic transaminase; GPT = Glutamic-pyruvic transaminase; Na = Sodium; K = Potassium; Ca = Calcium; P = Phosphorus; Mg = Magnesium; Vit B12 = Vitamin B12; Vit D = Vitamin D; ECG = Electrocardiogram; MOC = Computerized Bone Mineralometry; DEXA = Dual Energy X-ray Absorptiometry.

**Table 4 nutrients-13-02307-t004:** Side effects of ketogenic diet. * Not directly observed by the authors in the patients they followed on the ketogenic diet.

SIDE EFFECTS		
COMMON	INFREQUENT	VERY INFREQUENT
Muscle cramps Fatigue Hypotension Constipation Undesired weight loss	Hiperlipidaemia Gallbladder stones Menstrual irregularity Alopecia Nail fragility	Nausea Vomiting Abdominal pain Diarrhoea Prurigo pigmentosa (keto rash) * Mood disorders *

* Not directly observed by the authors in the patients they followed on the ketogenic diet.

## Data Availability

Not applicable.

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
