# Peer review of "Applications of Ketogenic Diets in Patients with Headache: Clinical Recommendations"

_nutrients, 2021, doi:10.3390/nu13072307_

Round 1

Reviewer 1 Report

Comments to the Author

The manuscript by Di Lorenzo and colleagues reported whether application of ketogenic diets (KD) treats headache and migraine. The authors were tempted to translate so far available data into possible strategy for the treatment of migraine and other headaches.

Even though the idea behind this review is interesting and in general, nonpharmacological treatments including (KD) emerge as an alternative strategy with increasing momentum, the article is poorly written, and available data are presented in such a long, pedantic and annoying way concealing the essence of the data. Moreover, the introduction is too long presenting unnecessary facts to justify the rational of the review.

I have substantial comments that the authors should address:

  1. The various protocols of KD should be shortened.
  2. The authors explained different actions mechanisms of KD but failed to link them to migraine genesis.
  3. I wonder why not mentioning the role of ion channels in KD as increasing evidence implicates ion channels in migraine and associates KD to ion channels.
  4. The authors should consider providing an overview table emphasizing containing recommendations on the management of KD for headache patients.

Minor

Wrong sentences: for instance

The mechanisms of action by which the ketogenic and non-ketogenic nutritherapeutic approaches may act are multiple (Figure 4), summarized as follows. Page 7 line 238.

Author Response

Replies to Reviewer 1

The manuscript by Di Lorenzo and colleagues reported whether application of ketogenic diets (KD) treats headache and migraine. The authors were tempted to translate so far available data into possible strategy for the treatment of migraine and other headaches.

Even though the idea behind this review is interesting and in general, nonpharmacological treatments including (KD) emerge as an alternative strategy with increasing momentum, the article is poorly written, and available data are presented in such a long, pedantic and annoying way concealing the essence of the data. Moreover, the introduction is too long presenting unnecessary facts to justify the rational of the review.

R: We thank the reviewer for pointing out an important weakness in our manuscript. Thanks to your observation, we have had the text drastically revised to improve it.

I have substantial comments that the authors should address:

  1. The various protocols of KD should be shortened.

R: Thanks for the suggestion. We shortened the section.

2. The authors explained different actions mechanisms of KD but failed to link them to migraine genesis.

R: We agree with the reviewer’s observation and the earlier statement that the manuscript is too long. We have removed mention of less well-established mechanisms and added in statements regarding how specific mechanisms are relevant to migraine when appropriate.

About Cerebral energy metabolism, we added: “The availability of higher levels of ATP can improve the above-mentioned migraine-specific brain energetic deficit.”.

About the Mitochondrial Functioning, we added: “Mitochondrial dysfunctions has also been shown in patients with migraine, , and recovery of function correlated with an improvement in migraine symptoms [69,70]. In particular, migraine improves with high doses of riboflavin that acts in the recovery of deficiencies of the electron transport chain of mitochondrial Complex 1 [69] that also is a selective metabolic target of KD. The effect of ketones as mitochondrial boosted could account for their protective effect in migraine”.

About Oxidative Stress, we added: “oxidative stress has been hypothesized to be the common point among all the mechanisms of action by which the different migraine triggers induce the initiation of the migraine attack [72]. Indeed, the use of antioxidant nutraceutical compounds has been widely adopted by migraine patients [73] and exogenous ketones are among these products.”

About Epigenetics, we added: “BDNF was related to migraine susceptibility both in episodic[92] and chronic forms[93], maybe by the BDNF-induced pain related neural plasticity [94], and its modulation by BHB can improve the clinical picture”.

3. I wonder why not mentioning the role of ion channels in KD as increasing evidence implicates ion channels in migraine and associates KD to ion channels.

R: We are grateful to the reviewer for the valuable suggestion that enriched our discussion. We added the following paragraph and modified figure 4.

"Ion channels

One of the mechanisms of action proposed to explain the effect of ketogenic diet concerns the modulation of ion channels, in particular the adenosine triphosphate-sensitive potassium channels (KATP channels) that are opened by KD metabolites, reducing firing in central neurons [107]. KATP channels are being evaluated with interest as possible novel therapeutic targets for migraine because of their involvement in migraine pathophysiology [108]. On the other hand, KD has shown to be effective in alternating hemiplegia of childhood related to ATPase Na+/K+ Transporting Subunit Alpha 3 (ATP1A3) gene mutations. ATP1A3 gene mutations alter the ionic currents across the cell membrane and account for a wide spectrum of neurological disorders, including hemiplegic migraine [109]".

4. The authors should consider providing an overview table emphasizing containing recommendations on the management of KD for headache patients.

R: Tanks for the suggestion, see table 1

"An overview of recommendations is summarized in Table 1".

Minor

Wrong sentences: for instance

The mechanisms of action by which the ketogenic and non-ketogenic nutritherapeutic approaches may act are multiple (Figure 4), summarized as follows. Page 7 line 238.

Thanks for the advice, we rephrased the sentence: “Ketogenic diets have been shown to have multiple mechanisms of action, many of which may be relevant in the management of headache (Figure 4)”.

Reviewer 2 Report

The primary merit of the paper is to initiate possibly a discussion and research on KD in migraine and other headaches. 
There are several short-comings:

Main concerns:

  • In the introduction "lifestyle changes" are only mentioned as alternative amongst others to the primary prophylactic tool medication. But behavioral therapy, lifestyle changes and relaxation therapies are the corner stone of every headache prophylaxis not only in children ans adolescents. This should be discussed in the introduction ans also in the patient selection part of the recommendations. 
  • It should clearly emphasized that the recommendations of the study group are based of personal clinical experience and not at all on clinical studies.
  • On the basis of up-to-date data, it is not possible to assess the prospects of KD in headache disorders. This hat to be stated clearly. Nevertheless, it makes sense to use it as individual treatment attempt.
  • While the capter on possible mechanisms of action cites many references the recommendations chapter doesn't refer to international recommendations regarding KD (e. g. Epilepsia Open, 3(2):175–192, 2018).

Further concerns:

  • The clearest indications for KD illustrate the metabolic function of ketones. They are not mentioned at all:
    Glucose transporter protein 1 (Glut-1) deficiency syndrome,
    Complex 1 mitochondrial disorders, and
    Pyruvate dehydrogenase deficiency.
  • The functional role of the so-called "migraine generator" is still completely unclear.  
  • From which age the authors recommend to think about KD.
  • Are there no preferences of the study group which type of KD should be used? 
  • Table 2 has to be reformated regarding the line breaks. I would recommend to list investigations and not pathologic conditions (as "Azotemia"...). It might be helpful to look at the tables 4 and 6 of Epilepsia Open, 3(2):175–192, 2018.
    BMD/DEXA is not standard in KD monitoring.    
  • 5.7: Duration of the diet:
    It would be interesting to know after which time the effect was seen in the studies or reports cited in 3.1, and also after which time repsonse was seen in studies on epilepsy (e. g. Dravet, CSWS, etc.). 

Author Response

The primary merit of the paper is to initiate possibly a discussion and research on KD in migraine and other headaches.
There are several short-comings:

Main concerns:

  • In the introduction "lifestyle changes" are only mentioned as alternative amongst others to the primary prophylactic tool medication. But behavioral therapy, lifestyle changes and relaxation therapies are the corner stone of every headache prophylaxis not only in children and adolescents. This should be discussed in the introduction and also in the patient selection part of the recommendations.

R: We completely agree with the reviewer and following the suggestion, we rephrased the core of the section in this way: “American Headache Association guidelines [10] recommend behavioral therapy, relaxation therapy [11], and other lifestyle changes [12,13] as the cornerstones of headache prophylaxis and recommend discussing these among the treatment options with all patients [10]. Among the NPAs, other than relaxation and lifestyle changes, the greatest interest is in nutraceuticals [14], and nutritional interventions [15], including food selection (in terms of potential food trigger avoidance) [16,17], and potential dietary treatment [18,19].”

In the patient selection part of the recommendations, we rephrased and added: “The expert panel agreed that patients with migraine appropriate for referral for KD, as complementary and supportive to other NPAs, includes…”.

  • It should clearly emphasize that the recommendations of the study group are based of personal clinical experience and not at all on clinical studies.

R: We agree with reviewer. To emphasize it we added 2 sentences. At the end of introduction, we clearly stated that: “The summary of our study group recommendations represents expert opinions in the field of headache management, based on members’ personal clinical experiences, in instances where lacking rigorous scientific studies to create evidence-based guidelines”. In the future prospective paragraph, we added the sentence: “We anticipate that future revisions of these recommendations will be based not on expert opinion but on evidence from rigorous clinical trials”.

  • On the basis of up-to-date data, it is not possible to assess the prospects of KD in headache disorders. This hat to be stated clearly. Nevertheless, it makes sense to use it as individual treatment attempt.

R: Also in this case, we agree with the reviewer suggestion. At the beginning of recommendations chapter we added the sentence: “Based on up-to-date data, it is not possible to assess all of the possible applications of KD in headache disorders. Nevertheless, the above-mentioned evidence suggests that it may be beneficial in a variety of headache disorders”.

  • While the chapter on possible mechanisms of action cites many references the recommendations chapter doesn't refer to international recommendations regarding KD (e. g. Epilepsia Open, 3(2):175–192, 2018).

R: Thanks, we added the reference in the KD overview and in recommendation sections.

Further concerns:

  • The clearest indications for KD illustrate the metabolic function of ketones. They are not mentioned at all:
    Glucose transporter protein 1 (Glut-1) deficiency syndrome,
    Complex 1 mitochondrial disorders, and
    Pyruvate dehydrogenase deficiency.

R: Thanks for the suggestion. In the overview chapter, we added: “In addition, KD is presently considered the only therapy in the following metabolic disorders: Glucose transporter protein 1 (GLUT-1) deficiency syndrome, Complex 1 mitochondrial disorders (C1MDs), and Pyruvate dehydrogenase deficiency [30]".

Moreover, we added in the Chapter of Mechanisms of Action, in the paragraph of Mitochondial functioning, we added these two sentences: “Moreover, KD has been shown to restore mitochondrial Complex 1 function in C1MDs [65]” and “Mitochondrial dysfunctions has also been shown in patients with migraine, and recovery of function correlated with an improvement in migraine symptoms [69,70]. In particular, migraine improves with high doses of riboflavin that acts in the recovery of deficiencies of the electron transport chain of mitochondrial Complex 1 [69] that also is a selective metabolic target of KD. The effect of ketones as mitochondrial boosted could account for their protective effect in migraine”.

  • The functional role of the so-called "migraine generator" is still completely unclear. 

R: the reviewer has touched a nerve on a debate in migraine pathophysiology that is very difficult to face in few lines. Anyway, we explained that “from 24 hours before the spontaneous migraine attack, hypothalamic activity is increased and altered in functional coupling between the spinal trigeminal nuclei, but with the dorsal rostral pons during the attack”.

  • From which age the authors recommend to think about KD.

R: The reviewer effectively identified a limitation of our study group. All of our board members treat adult patients, so we do not have specific experience in the pediatric setting. We added a section about younger subjects in the chapter of clinical evidence in migraine:

“Few studies with negative results are available examining younger patients with headache. In particular, it was observed the lack of effect of MAD in adolescents with CDH [45]. This finding was not unexpected because of specific diagnostic and therapeutic challenges present in pediatric patients with headache. Instead, according to the ICHD-3, the interpretation of pediatric headache, and response to treatment, may not be clear, either because of the child's difficulty in reliably reporting symptoms or because of their milder manifestations in pediatric migraine [2], therefore, misdiagnosis and misinterpretation are always possible”.

In the chapter of recommendations, we disclose our lack of experience in this setting:

“Although none of the board members has extensive experience treating children and adolescents with diet therapies for headache, the large body of scientific literature available on the efficacy of KD in neurological disorders in children and reviewed in this manuscript support potential application in early life. However, more studies evaluating safety, feasibility, and efficacy of ketogenic diet therapies in the management of headache in children and adolescents are needed [148]”.

  • Are there no preferences of the study group which type of KD should be used?

R: Thanks for the highlighting of this aspect that was widely discussed during the work of our board. In the recommendation chapter, Multidisciplinary evaluation and diet therapy paragraph, we added the sentence “The board concluded that there is insufficient evidence to guide the selection of the best type of diet to attempt first, both because each group utilized different diets or diet combination, and because the decision was left primarily to the nutrition professional, considering the preferences of the individual patient”.

  • Table 2 has to be reformated regarding the line breaks. I would recommend to list investigations and not pathologic conditions (as "Azotemia"...). It might be helpful to look at the tables 4 and 6 of Epilepsia Open, 3(2):175–192, 2018.
    BMD/DEXA is not standard in KD monitoring.

R: We have modified the table, as suggested.

Even if not standard in pediatric population, the board found necessary to periodically assess bone density for specific sex (mainly females) and age risk to develop osteoporosis.

  • 5.7: Duration of the diet:
    It would be interesting to know after which time the effect was seen in the studies or reports cited in 3.1, and also after which time repsonse was seen in studies on epilepsy (e. g. Dravet, CSWS, etc.).

R: Few data are available about it, but is shared opinion, supported by all the reports we have cited in the text, that usually it is very quick.

“Based on the clinical experience of the board members, patients’ response to KD is typically very quick, generally within one week. However, sometimes improvement is more delayed. If there is no response to the diet within 3 months, increasing the ketogenic ratio could be considered. In patients with a of lack of response in the presence of MOH, continuation of the diet while discontinuing the drug responsible for MOH without corticosteroids may be effective. If lack of efficacy persists, the diet can be discontinued by month 6”.

Reviewer 3 Report

Excellent review of the ketogenic diet’s use in migraines, and the possible mechanisms. The recommendations section is very thorough. I see no need for revisions.

Author Response

Thanks, we would like to thank the reviewer for appreciating our manuscript.

Round 2

Reviewer 2 Report

The text has improved considerably in the revised version. In the present form, it is a valuable overview.

Author Response

Thanks for your review that leads us to improve the manuscript.